# Lime Rate in Clayey Soils Influences Chemical Fertility and Sugarcane Yield

**DOI:** 10.3390/plants11162110

**Published:** 2022-08-13

**Authors:** Murilo de Campos, Jorge Martinelli Martello, Gabriela Ferraz de Siqueira, Ariani Garcia, Daniele Scudeletti, Patrícia Pereira Dias, Raffaella Rossetto, Juliano Carlos Calonego, Heitor Cantarella, Carlos Alexandre Costa Crusciol

**Affiliations:** 1Lageado Experimental Farm, Department of Crop Science, College of Agricultural Sciences, São Paulo State University (UNESP), Botucatu 18610-307, SP, Brazil; 2São Paulo State Agency for Agribusiness Technology, Piracicaba 13400-970, SP, Brazil; 3Soils and Environmental Resources Center, Agronomic Institute (IAC), Campinas 13075-630, SP, Brazil

**Keywords:** conventional tillage, deep strip-tillage, furrower, localized tillage

## Abstract

Liming contributes to the alleviation of acidity in highly weathered soils. For sugarcane, the use of green harvest methods and new soil tillage systems requires an adjustment of lime application rates. In the present study, the effects of different lime rates and tillage systems on sugarcane performance and soil chemical fertility parameters were assessed. Three experiments were conducted in two locations between April 2015 and October 2019. The study design was a randomized block field study with four replicates. Four lime rates were applied once at sugarcane establishments in each soil tillage system and location: no liming (control); lime recommended rate (LRR); two times LRR (2× LRR); and three times LRR (3× LRR). The three soil tillage systems were conventional (CT), deep-strip (DT), and modified deep-strip tillage (MDT). Soil chemical fertility, leaf nutrient concentrations, and sugarcane yield components were analyzed, and correlations were identified by principal component analysis (PCA). The soil acidity was adequately alleviated in all tillage systems. Increasing the lime rate improved the lime distribution and soil fertility parameters. Applying lime at rates higher than LRR improved stalk and sugar yields, longevity, agronomic efficiency index (AEI), and correlated with a longer residual effect of liming, mainly in the last ratoon.

## 1. Introduction

Brazil is recognized as the biggest producer of sugarcane (*Saccharum* spp.) in the world [1], and Brazilian research has contributed greatly to advances in agronomic management. In the 2021/2022 growing season, sugarcane stalk production of approximately 592 million tons is expected in Brazil [2]. Despite this high production, the average yield in Brazil is approximately 71.2 Mg ha^−1^ [2], far below the potential of up to 200 Mg ha^−1^ [3]. This gap highlights the need to evaluate new practices that can increase the efficiency and profitability of the sugar and energy industries. Improving yield efficiency while conserving the soil is an intrinsic concern in modern agriculture, and sugarcane cultivation is no exception. Sugarcane is a semi-perennial crop, and successive cuttings are accompanied by progressively declining stalk and sugar yields, mainly due to reduced soil fertility, soil compaction, and ratoon damage caused by machinery traffic [4].

Sugarcane cultivation is preferentially localized in humid tropical regions where the soil is mainly highly weathered and acidic [5,6]. Consequently, agronomic management to alleviate soil acidity is indispensable for sugarcane cultivation. Agricultural lime is the most commonly used soil acidity amendment, and its effects are well documented in the literature [7,8,9]. In short, lime has a double purpose as a source of calcium (Ca) and magnesium (Mg) and as a soil acidity amendment. Ca plays a fundamental role in cell wall structure and membrane formation [10,11], whereas Mg participates in all energy-dependent physiological processes and photosynthesis [12,13]. Carbonate hydrolysis of lime generates hydroxyl (OH^−^) ions capable of reacting with hydrogen (H^+^) ions, thereby increasing soil pH and effective cation exchange capacity (ECEC) [14].

For sugarcane, lime application is based on guidelines from regional bulletins calibrated to conventional soil tillage and burned-harvest systems [15]. Although conventional soil tillage remains the primary form of sugarcane tillage management, the advent of GPS-controlled machinery traffic has opened new possibilities for soil tillage systems, including localized tillage management [16,17]. In addition, green harvest methods, i.e., harvest without burning, are gaining ground in Brazil. These methods improve soil quality and structure by increasing soil organic matter (SOM), cycling, and storage capacity [18] but also necessitate readjustment of lime rates and forms of application depending on the tillage system adopted.

Under conventional soil tillage, lime incorporation is limited to the mobilized layer (approximately 0.4 m). Since the sugarcane root system can reach deeper layers [19], superficial lime incorporation might be ineffective in highly acidic subsurface soils. To enable deeper lime placement, sugarcane producers are increasingly adopting localized tillage management, such as deep strip-tillage on planting furrows, reserving inter-rows specifically for machinery traffic. However, the few available studies have primarily focused on soil physical evaluations and operational efficiency [16,17,20,21,22,23,24] rather than soil chemical fertility and soil acidity management. Addressing this gap is important, as chemical impediments can reduce the exploration of deeper layers by the plant root system, particularly in acidic soils [25,26].

We hypothesize that higher-than recommended lime doses will enhance soil chemical fertility under deep strip-tillage due to the increase in vertical soil mobilization, which improves the arrangement and distribution of lime. Moreover, we hypothesize that, regardless of the tillage system, higher-than-recommended lime doses will contribute to acidity amendment long-term (residual effect), as reflected in increased stalk and sugar yields in clayey, acidic, and highly weathered soils. To test these hypotheses, in this study we evaluated the impact of increasing lime rates on soil acidity amendment and soil fertility throughout the soil profile and the yield of stalks and sugar under conventional, deep strip-tillage, and modified deep strip-tillage for the complete sugarcane growing cycle.

## 2. Results

Although differences were observed among the tillage systems, in general, lime application buffered soil pH efficiently and maintained acidity levels ranging from moderate (5.0–5.5) to slight elevated (>5.5) [27] in plant cane and first ratoon. The rates 2 and 3× LRR buffered the soil pH more efficiently than LRR in second and third ratoons, especially at depths ≤ 0.6 m at both locations (Appendix A).The soil concentrations of Ca^2+^ and Mg^2+^ (Appendix A) and the residual effect of liming were highest under MDT. The liming process distributed equivalent amounts of Ca and Mg under CT and DT but not MDT. Under CT and DT, the Ca and Mg inputs were 1065 and 510 kg ha^−1^ for LRR, respectively, at Macatuba, and 850 and 550 kg ha^−1^ for LRR, respectively, at Piraju. Under MDT, the same rate calculated for the total area was applied as concentrated bands. Given a row spacing of 1.5 × 0.9 m (i.e., 2.4 m) and a furrower working width of 1.2 m, a double dose was strip incorporated by the furrower. Accordingly, under MDT, the Ca and Mg inputs were 2130 and 1020 kg ha^−1^ for LRR, respectively, at Macatuba and 1700 and 1100 kg ha^−1^, respectively, at Piraju.

Because base saturation (BS) is highly influenced by Ca^2+^ and Mg^2+^, the effects of the treatments on BS reflected the inputs of Ca and Mg. Under CT at Macatuba, BS was significantly higher under liming than in the control at depths ≤ 0.6 m under LRR and in all layers under 2× and 3× LRR, except in third ratoon, in which significant effects were observed only at depths ≤ 0.6 m (Appendix A). In Piraju, BS increased with the lime rate in the first layer (0.0–0.2 m) and at all depths in plant cane, at depths ≤ 0.6 m in first and second ratoons, and depths ≤ 0.4 m in third ratoon (Appendix A).

Under DT at Macatuba, BS was higher in all limed treatments than in the control at all depths in plant cane, first and second ratoons and at depths ≤ 0.4 m in third ratoon (Appendix A). At Piraju, lime application increased BS at depths ≤ 0.4 m in all growing seasons, regardless of the lime rate. At depths of 0.4 to 0.6 m, all rates of lime application increased BS in plant cane, whereas only 2× and 3× LRR increased BS in first ratoon, and only 3× LRR increased BS in second and third ratoons (Appendix A).

Compared to the control, BS was higher in the limed treatments under MDT at all lime rates, soil depths, and growing seasons, except at depths ≥ 0.8 m in third ratoon. In general, at Macatuba, BS increased with the lime rate, even in the most subsurface layers (Appendix A). At Piraju, liming significantly increased BS at depths ≤ 0.6 m regardless of the lime rate and growing season. In third ratoon, only 2× and 3× LRR had significant effects from 0.2 to 0.6 m (Appendix A).

BS above 60% is considered adequate for sugarcane according to the soil fertility recommendation bulletin of São Paulo State [27]. As the LRR was calculated by summing the lime recommendations for the uppermost surface layers (0.0–0.2 + 0.2–0.4 m), BS values reaching 60% at depths of at least 0.4 m were expected. In general, adequate results were observed in plant cane under all tillage systems when LRR was used; however, in subsequent growing seasons, BS was less than 60% under LRR even in the first layer, regardless of the tillage system. When higher lime rates were used, a greater residual effect was observed, mainly in first ratoon under CT and DT and in all ratoons under MDT. BS varied between the sites and was highest in the first layer. AS was significantly reduced by liming under CT regardless of rate at all depths in plant cane and first ratoon. In second and third ratoons, liming under CT increased AS at depths ≤ 0.6 m at Macatuba (Appendix A) and Piraju (Appendix A). Under DT, the effect of liming on AS was evident throughout the entire soil profile in plant cane and first ratoon and at depths ≤ 0.8 m in second ratoon, regardless of the rate. In third ratoon, liming reduced AS at depths ≤ 0.4 m at Macatuba (Appendix A) and ≤ 0.6 m at Piraju (Appendix A). Under MDT, liming significantly reduced AS at all rates and soil depths in plant cane, first and second ratoons and at depths ≤ 0.6 m in third ratoon at Macatuba (Appendix A). At Piraju under MDT, AS was controlled by liming regardless of rate and depths in plant cane and at depths ≤ 0.6 m in all ratoons (Appendix A).

The leaf concentrations of Ca and Mg increased with the lime rate at both locations but not in all growing seasons (Appendix A). In general, there was no clear pattern of significant changes in the leaf concentrations of elements in any of the growing seasons, but on average, none of the values was outside the ranges proposed by ref. [28] for sugarcane (Appendix A). Thus, the plants were considered well nourished regardless of treatment and tillage system.

Principal component analysis (PCA) was performed to identify the parameters explaining the variations in stalk and sugar yields. The PCA considered BS, AS, soil concentrations of Ca^2+^ and Mg^2+^, and soil depth as parameters (Figure 1, Figure 2 and Figure 3). Regardless of the tillage system, eigenvalues ≥1 were present only in factor 1 (horizontal axis). In factor 2 (vertical axis), all eigenvalues were <1, and thus this factor was not considered (Appendix A). The PCA revealed positive influences of Ca^2+^, Mg^2+^, and BS and negative influences of AS on stalk and sugar yields for both tillage systems and locations.

Under CT, correlations of BS and AS with stalk and sugar yields ≥ |0.70| were observed in all soil layers and locations, whereas Ca^2+^ and Mg^2+^ were correlated with stalk and sugar yields at depths ≤ 0.6 and 0.4 m, respectively, in Macatuba and at depths ≤ 0.4 and 0.6 m, respectively, in Piraju (Figure 1a,c; Appendix A).

Under DT, stalk and sugar yields were negatively correlated with AS in all soil layers in Macatuba and at depths ≤ 0.8 m in Piraju. For BS, Ca^2+^, and Mg^2+^, significant correlations with stalk and sugar yields were observed at depths ≤ 0.8, 0.6, and 0.4 m, respectively, in Macatuba and depths ≤ 0.4 m in Piraju (Figure 2a,c; Appendix A).

Under MDT, BS and AS were significantly correlated with stalk and sugar yields in all soil layers, whereas Ca^2+^ and Mg^2+^ were correlated with stalk and sugar yields at depths ≤ 0.4 and 0.8 m, respectively, in Macatuba. In Piraju, significant effects on stalk and sugar yields were observed at depths ≤ 0.6 m for BS and Ca^2+^, ≤ 0.8 m for AS, and ≤0.4 m for Mg^2+^ (Figure 3a,c; Appendix A).

Under all tillage systems at both locations, all lime rates in the two first growing seasons, 2× and 3× LRR in second ratoon, and 3× LRR in the last growing season were associated with the yield parameters, except for the last growing season under DT at Piraju (Figure 1b,d, Figure 2b,d, and Figure 3b,d).

The absolute results also demonstrated the dependence of stalk and sugar yields on higher-than-recommended lime rates, regardless of the tillage system. In Macatuba, when all growing seasons were averaged, applying 2× and 3× LRR increased the stalk yield by 13.4 and 18.6 Mg ha^−1^ under CT, 12.0 and 22.4 Mg ha^−1^ under DT, and 7.0 and 8.8 Mg ha^−1^ under MDT. The application of 2× and 3× LRR increased the sugar yield by 2.1 and 2.8 Mg ha^−1^ under CT, 2.0 and 3.4 Mg ha^−1^ under DT, and 1.0 and 1.5 Mg ha^−1^ under MDT. When all growing seasons in Piraju were averaged, 2× and 3× LRR increased the stalk yield by 5.7 and 12.9 Mg ha^−1^ under CT, 6.0 and 14.5 Mg ha^−1^ under DT, and 10.7 and 22.0 Mg ha^−1^ under MDT. Under 2× and 3× LRR, sugar yield increased by 1.4 and 2.7 Mg ha^−1^ under CT, 1.3 and 2.6 Mg ha^−1^ under DT, and 1.9 and 4.2 Mg ha^−1^ under MDT. The sugar yield increased with the lime rate because it is based on stalk yield; however, in general, the sucrose concentration and TRS increased significantly when lime was applied regardless of the rate (Figure 4, Figure 5 and Figure 6).

The agronomic efficiency index (AEI) was calculated to evaluate the efficiency of 2× and 3× LRR compared with LRR. In all cases, the AEI values of 2× and 3× LRR were above 100%, confirming the greater efficiency of higher-than-recommended lime rates compared with LRR. Under CT and DT, across the complete sugarcane cycle (summing the growing seasons), the average AEI values indicated that 2× and 3× LRR were 73 to 149% more efficient than LRR in terms of stalk and sugar yields at Macatuba. At Piraju, the average AEI values for 2× and 3× LRR indicated approximately 20 to 50% greater efficiency. Under MDT, the average AEI values showed that the increase in stalk and sugar yield efficiency provided by 2× and 3× LRR was 24 to 35% at Macatuba and 31 to 70% at Piraju (Table 1).

## 3. Discussion

In general, although the magnitudes of the effects varied, increasing the lime rate increased soil Ca^2+^ and Mg^2+^ concentrations and BS and decreased AS. These effects were observed at both sites but varied in intensity according to the lime rate and tillage system. The highest rates of liming reached deeper soil layers efficiently in plant cane and intermediate soil layers in subsequent ratoons. These findings are important since the distribution of plant roots tends to be concentrated in soil layers with higher fertility levels and high water availability [7,28].

Applying higher-than-recommended lime rates provided large inputs of Ca and Mg, but the concentrations of exchangeable Ca and Mg did not increase proportionally with the lime rate because lime reactivity is pH dependent [29]. As liming increases the soil pH, the reactivity of the lime decreases, contributing to downward or lateral displacement of the unreacted lime particles. When acidity is restored (below approximately 4.6 to 5.0 (CaCl_2_)), the reactivity of the insoluble lime particles recovers [29,30], creating residual effects of liming. For perennial or semi-perennial crops, a greater residual effect of liming is desirable to counteract the progressive decline in soil fertility [4] and to avoid necrosis of root tips caused by Ca deficiency, which will compromise the efficient uptake of water and nutrients [31].

AS and BS are inversely proportional and thus equally important. Al species are highly pH dependent, and applying lime increases soil pH and decreases AS by complexing Al^3+^ into Al-hydroxide species via hydrolysis [32]. The efficiency of lime placement varied even in the deepest layers, and vertical movement of lime was also observed. In general, soil pH increased throughout the soil profile under all tillage systems, indicating that neutralization occurred beyond the lime application point. Mobilization of fine lime particles, whether in association with organic compounds or not, can occur by chemical and physical mechanisms, such as the formation of an alkalinization front and displacement by water movement through the openings left by the roots in the soil [26,33].

Lime application positively or negatively impacts the availability of most elements [34]. Liming increased the soil and leaf concentrations of Ca and Mg not only as a direct source of these nutrients but also by increasing the effective cation exchange capacity (ECEC), which promotes the deprotonation of mineral and organic particles and, consequently, electrostatic adsorption of cations [33]. This phenomenon was clearly evidenced by the higher BS when lime was applied, especially at the highest rate.

Conversely, cationic micronutrients may be specifically adsorbed by soil organic matter (SOM) or CEC when the pH increases, and the formation of low-solubility species can decrease their availability [35]. In this study, the soil was mobilized and thoroughly mixed with the lime at tillage, and micronutrient fertilization was adequate at sugarcane establishment; consequently, higher-than-recommended lime rates did not have any important negative effects on cationic micronutrients, which remained within the recommended ranges for sugarcane [27].

Compared with the other analyzed parameters, the effects of lime application on sugarcane quality parameters were smaller. Lime application generally increased the sucrose concentration and TRS, regardless of the rate. The improvements in soil chemical conditions and Mg^2+^ availability provided by liming are primarily responsible for these increases in sugarcane quality parameters, as there is a direct relationship between the biosynthesis and transport of sucrose and Mg availability [36]. In addition, the climate conditions during the study were beneficial, with well-distributed periods of rainfall, and the hydric balance was positive at the full vegetative growth stage at both locations, which may have favored all treatments and decreased their response to liming.

The attributes that best explained the variations in stalk and sugar yields were identified by PCA. BS (positively) and AS (negatively) were highly correlated with stalk and sugar yields at all soil depths, highlighting the importance of adequate soil fertility in deeper layers. The PCA also revealed that the increases in stalk and sugar yields under higher-than-recommended lime rates were greatest after the first ratoon, implying a longer lasting residual effect of the highest lime rates.

The response to increasing lime rate appeared to be greater under CT and DT, as the AEI values of the highest lime rates were higher under CT and DT than under MDT, especially in Macatuba. Under CT and DT, lime was broadcast over the entire area and better distributed throughout the soil profile under the highest rates. The greater contact with soil in these systems enhanced the dissolution capacity of the lime particles [37,38]. By contrast, under MDT, LRR was sufficient because the lime was applied in concentrated bands targeted directly at the planting furrows; consequently, the effects of higher lime rates on yields were less pronounced. In Piraju, the greater clay content seemed to buffer the soil pH more efficiently, and the highest AEI was reached under MDT with the highest lime rates.

The cumulative stalk and sugar yields after four growing seasons confirmed the improvement in sugarcane performance under the highest lime rates. This improvement was primarily attributable to efficient soil acidity alleviation, mainly at depths ≤ 0.6 m, and long-term residual effects in these soils with high buffering capacity.

## 4. Materials and Methods

### 4.1. Location, Soil, and Climatic Characterization of Experimental Areas

Three experiments in two sugarcane commercial sites were simultaneously carried out in Macatuba (Zilor group: 22°29′ S and 48°47′ W, 515 m of elevation) and in Pirajú (Raízen group: 23°07’ S 49°24’ W, 645 m of elevation), State of São Paulo, Brazil (Appendix A). The field trials were conducted for four harvests, from the plant cane cycle plus three regrowth (from April 2015 to August 2019). According to the Köppen classification, the predominant climate for both locations is tropical humid with a hot summer (Cwa). The average rainfall, temperature, and hydro-climatic balance [39,40] are presented in Appendix A. Both soils were classified as Rhodic Hapludox [41], and their chemical [42] and texture characterization [43] are presented in Appendix A.

### 4.2. Treatments and Experimental Design

Four lime rates (control, no liming; lime recommended rate (LRR) calculated by the base saturation (60%) method (sum of the soil layers from 0.0 to 0.2 and 0.2 to 0.4 m) [15]; two times LRR (2× LRR); and three times LRR (3× LRR)) were all applied at the sugarcane establishment using randomized block design with four replicates. Considering the most used soil tillage systems in sugarcane, conventional and deep-strip tillage were used. For the deep-strip tillage, two main patterns of lime broadcasting were chosen. Due to the complex dataset, each tillage system described below was individually analyzed only as a function of increasing lime rates.

A brief scheme of the soil tillage systems is given in Figure 7 and they were characterized by the operations: Conventional soil tillage system (CT)—this is the standard system with lime broadcast and incorporated (heavy offset disc harrow, subsoiler, and leveling disc harrow on 0.3–0.4 m) over the entire area (Figure 7a); deep strip-tillage system (DT)—the lime was broadcast over the entire area and incorporated into strips with a furrower (a rotary hoe with a working depth of 0.3 to 0.4 m and a subsoiling rod with a working depth and width of 0.7 m and 1.2 m, respectively) only at the planting rows (Figure 7b); modified deep strip-tillage system (MDT)—¾ of the lime quantity was broadcast over the soil surface in localized bands, and ¼ was distributed into the soil profile at 0.4 and 0.6 m through the subsoiling rod (lime storage compartment linked to the subsoiling rod) (Figure 7c). The term “modified” indicates the lime was broadcast concentrated in localized bands and further strip incorporated by the furrower.

The LRR for Macatuba (CaO and MgO concentration of 30 and 17%, respectively; relative efficiency, 95%) and Piraju (CaO and MgO concentration of 28 and 23%, respectively; relative efficiency, 82%) were evaluated considering a proportional relative efficiency of 100% and the respective lime doses for both sites and tillage systems are in Table 2.

The sugarcane varieties CTC 20 in Macatuba and CTC 11 in Piraju were mechanically planted (18 viable buds m^−1^ at about 0.3–0.4 m depth) immediately after liming using spacing of 1.50 × 0.90 m (double-combined row) and traffic system control. The experimental plots (20 m in length × 12 m in width) were constituted by 5 double rows per plot. The base fertilization consisted of applying 500 kg ha^−1^ of an NPK formula 06-28-24 in Macatuba and 600 kg ha^−1^ of an NPK formula 10-25-25 in Piraju. Additionally, 40 kg ha^−1^ of S were applied as elemental sulfur pastilles also at sugarcane establishment. The micronutrients zinc (Zn, 10%), boron (B, 1.8%), molybdenum (Mo, 0.12%), copper (Cu, 0.6%), and manganese (Mn, 1.2%) were applied separately in fillets 60 days after planting at a dose of 200 kg ha^−1^ The regrowths (ratoons) were managed by topdressing applications of 650 kg ha^−1^ of an NPK formula 18-07-28 in Macatuba and 600 kg ha^−1^ of an NPK formula 20-05-28 in Piraju. Both sites also received 40 kg ha^−1^ of S as elemental sulfur pastilles. The ratoons did not receive the lime application and other agricultural management practices were performed according to the sugar mill protocols.

### 4.3. Soil Chemical Analysis

The soil was sampled using a two-inch tubular sampler immediately after the sugarcane harvests, at depths from 0.0 to 1.00 m, stratified on each 0.2 m only considering the sugarcane planting furrows. The soil samples were air-dried and passed through a 10 mesh (2-mm) sieve. The pH (0.01 M CaCl_2_; 1:2.5, soil:solution) was determined using a previously calibrated FiveEasy Mettler Toledo potentiometer. The exchangeable calcium (Ca^2+^), magnesium (Mg^2+^), and potassium (K^+^) were extracted by ionic exchange resin and determined by atomic absorption spectroscopy (Shimadzu 9300) and by flame photometry (PFP7 model, Jenway, Staffordshire, UK), respectively. The exchangeable aluminum (Al^3+^) (extracted by 1 mol L^−1^ KCl solution) and potential acidity (H + Al) (extracted with 1 mol L^−1^ calcium acetate solution) were both determined by titration with ammonium hydroxide (0.025 mol L^−1^ [42]. The soil fertility parameters were calculated as follows: (1) base saturation (BS,%) = (SB/CECp) × 100, in which SB is the sum of cations that do not hydrolyze under normal soil pH conditions (Ca^2+^ + Mg^2+^ + K^+^) and the CEC_p_ is the potential cation exchange capacity (CEC_p_ = (SB + (Al+H))); (2) aluminum saturation (AS,%) = (Al^3+^/CEC_e_) × 100. in which the CEC_e_ is the effective cation exchange capacity (CEC_e_ += BS + Al^3+^).

### 4.4. Foliar Diagnosis

Ten leaves (top visible dewlap) were sampled in the two central rows at full vegetative growth collecting the middle third length. The plant material was dried (60 °C until constant mass), milled, and the concentrations of N, P, K, Ca, Mg, S, Cu, Fe, Mn, Zn, and B were determined [44].

### 4.5. Sugarcane Stalk and Sugar Yields and Agronomic Efficiency Index (AEI)

The stalks (topped at apical bud height and defoliated) were weighed in four sequential meters at two central rows and the values were extrapolated to hectares calculating stalk yield (Mg ha^−1^). After that, the cleaned stalks were processed to determine the apparent percentage of sucrose by polarization and total recoverable sugar (TRS, kg of sugar per Mg of stalk) [45,46]. The sugar yield was estimated by: (3) Sugar yield (Mg ha^−1^) = (TRS × Stalk yield/1000. After collecting the samples for stalks and sugar yield, the sugarcane was mechanically harvested (non-burning) in both sites (Macatuba: plant cane, first, second, and third ratoons in October 2016, October 2017, October 2018, and September 2019, respectively; and Piraju: plant cane, first, second, and third ratoons in December 2016, December 2017, December 2018, and November 2019, respectively).

The agronomic efficiency index (AEI) by the use of higher-than-recommended rates of lime was calculated by the ratio of stalk and sugar yields applied at the same lime rate for each growing season, as follows: (4) AEI (%) = [(Y2 − Y1)/(Y3 − Y1)] × 100, where Y1 = sugarcane stalks or sugar yield in the treatment control; Y2 = sugarcane stalks or sugar yield using 2 or 3× LRR; and Y3 = sugarcane stalks or sugar yield using LRR. The AEI determined the increments of stalk and sugar yields compared with the control (increased yields) and considered the ratio of the increment under 2 or 3× LRR to the increment under LRR, expressed as a percentage. Accordingly, values above 100% indicate that the higher lime rate was more efficient than LRR, while values below 100% indicate that LRR was more efficient than the higher lime rate. Values near or equal to 100% indicate no difference in efficiency between the lime rates.

### 4.6. Statistical Analysis

Data were submitted for analysis of variance (ANOVA) with subsequent comparison of means. Fisher’s test, supported to the least significant difference (LSD), was performed comparing soil chemical and fertility attributes for each soil depth, foliar diagnosis, and yield components in the same growing season. The 0.1 level of probability was used at all tests.

To identify that the major soil attributes and the sampled depths correlated with sugarcane stalks and sugar yield, the principal component analysis (PCA) technique was constructed separately for each tillage system and location using Pearson’s correlation coefficients (*p* ≤ 0.1) and determined through the Kaiser rule, considering eigenvalues >1 or explaining over 85% of the total variance [47]. Correlations > |0.70| were considered based on the number of paired values [48]. The average values of all growing seasons were used for both active (stalk and sugar yield) and supplemental (soil attributes at the corresponding depths) factors in PCA. All tests were performed by using the statistical software package from Statistica [49].

## 5. Conclusions

The present study assessed the effects of increasing lime rates on chemical fertility parameters throughout the soil profile and sugarcane performance during the complete sugarcane growing cycle under different tillage systems. In general, regardless of the tillage system, soil acidity was adequately alleviated, and increasing the lime rate improved the distribution of lime in the soil profile and soil fertility. The highest lime rates provided more pronounced residual effects.

Because micronutrient fertilization was adequate, higher-than-recommended lime rates did not negatively impact the nutritional status of the plant. Stalk and sugar yields and the AEI were positively impacted by higher-than-recommended rates, but sucrose concentration and TRS were less responsive to the lime rate, regardless of the tillage system.

Positive correlations of the yield parameters with soil Ca^2+^ and Mg^2+^ concentrations and BS were observed in PCA, mainly at depths ≤ 0.6 m under DT and at all soil depths under CT and MDT. Conversely, AS was negatively correlated with stalk and sugar yields at all depths, regardless of the tillage system. In addition, higher sugarcane yield and longevity were associated with a higher residual effect of liming, mainly in the last ratoon.

The stalk and sugar yields were highest at the higher-than-recommended lime rates, regardless of the tillage system and location. When all growing seasons were averaged, the AEI was highest under CT and DT in Macatuba and under MDT in Piraju.

This study provides solid information for chemical assessments of clayey soils, but similar studies in medium-textured and sandy soils are also necessary due to the widespread cultivation of such soils with sugarcane in Brazil.

## Figures and Tables

**Figure 1 plants-11-02110-f001:**
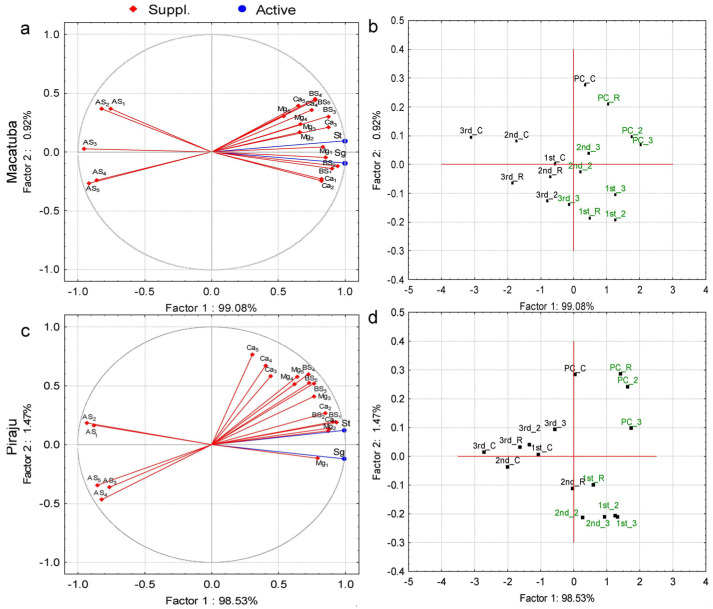
Projection of the dataset subjected to principal component analysis based on the correlations of stalk (St) and sugar (Sg) yields with (**a**,**c**) soil chemical attributes (Ca, calcium; Mg, magnesium; BS, base saturation; and AS, aluminum saturation) and depths (1, 0.0–0.2 m; 2, 0.2–0.4 m; 3, 0.4–0.6 m; 4, 0.6–0.8 m; and 5, 0.8–1.0 m); and (**b**,**d**) lime rate (C, control; R, recommended lime rate; 2, 2× recommended lime rate; 3, 3× recommended lime rate) at Macatuba and Piraju under conventional tillage (CT) for plant cane (PC) and first (1st), second (2nd), and third (3rd) ratoons.

**Figure 2 plants-11-02110-f002:**
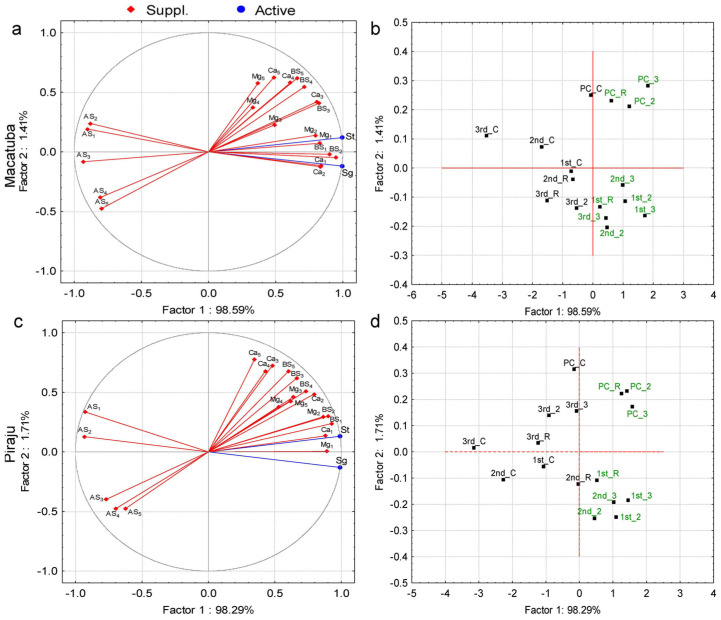
Projection of the dataset subjected to principal component analysis based on the correlations of stalk (St) and sugar (Sg) yields with (**a**,**c**) soil chemical attributes (Ca, calcium; Mg, magnesium; BS, base saturation; and AS, aluminum saturation) and depths (1, 0.0–0.2 m; 2, 0.2–0.4 m; 3, 0.4–0.6 m; 4, 0.6–0.8 m; and 5, 0.8–1.0 m); and (**b**,**d**) lime rate (C, control; R, recommended lime rate; 2, 2× recommended lime rate; 3, 3× recommended lime rate) at Macatuba and Piraju under deep strip-tillage (DT) for plant cane (PC) and first (1st), second (2nd), and third (3rd) ratoons.

**Figure 3 plants-11-02110-f003:**
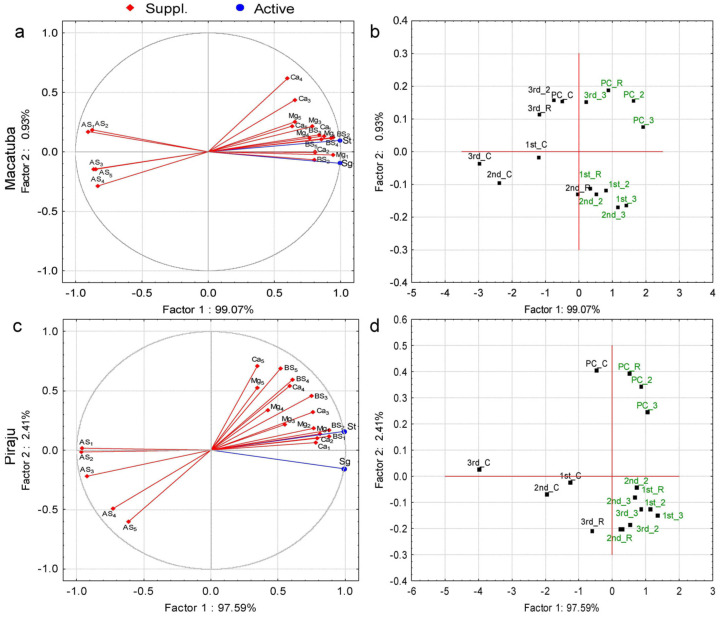
Projection of the dataset subjected to principal component analysis based on the correlations of stalk (St) and sugar (Sg) yields with (**a**,**c**) soil chemical attributes (Ca, calcium; Mg, magnesium; BS, base saturation; and AS, aluminum saturation) and depths (1, 0.0–0.2 m; 2, 0.2–0.4 m; 3, 0.4–0.6 m; 4, 0.6–0.8 m; and 5, 0.8–1.0 m); and (**b**,**d**) lime rate (C, control; R, recommended lime rate; 2, 2× recommended lime rate; 3, 3× recommended lime rate) at Macatuba and Piraju under modified deep strip-tillage (MDT) for plant cane (PC) and first (1st), second (2nd) and third (3rd) ratoons.

**Figure 4 plants-11-02110-f004:**
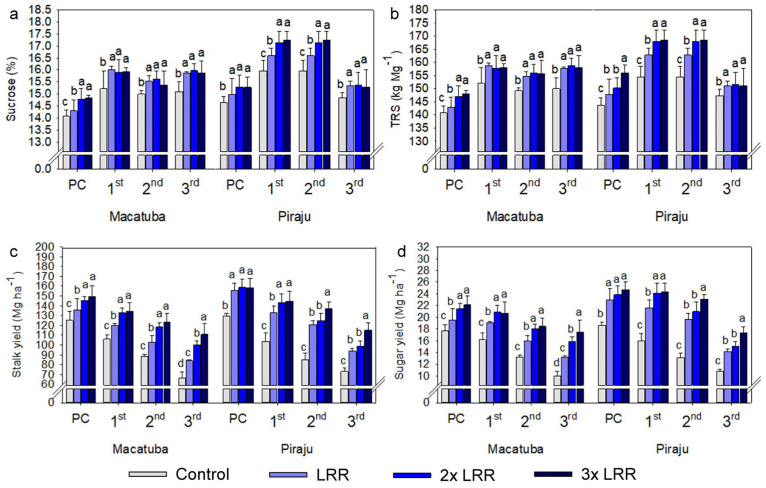
Sucrose (**a**), total recoverable sugar (TRS, **b**), and yields of stalks (**c**) and sugar (**d**) under increasing lime rates at Macatuba and Piraju under conventional tillage (CT) in plant cane and first, second and third ratoons. Different lowercase letters indicate significant differences between lime rates for each growing season (LSD, *p* ≤ 0.1). The error bars express the standard error of the mean (*n* = 4).

**Figure 5 plants-11-02110-f005:**
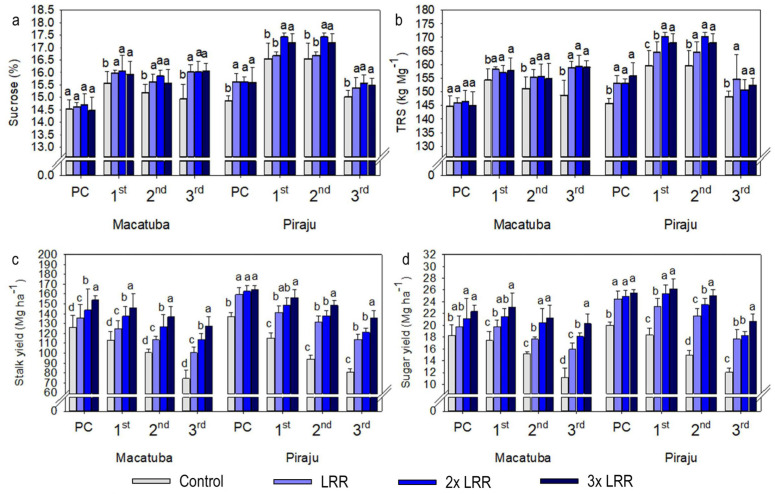
Sucrose (**a**), total recoverable sugar (TRS, **b**), and yields of stalks (**c**) and sugar (**d**) under increasing lime rates at Macatuba and Piraju under deep strip-tillage (DT) in plant cane and first, second, and third ratoons. Different lowercase letters indicate significant differences between lime rates for each growing season (LSD, *p* ≤ 0.1). The error bars express the standard error of the mean (*n* = 4).

**Figure 6 plants-11-02110-f006:**
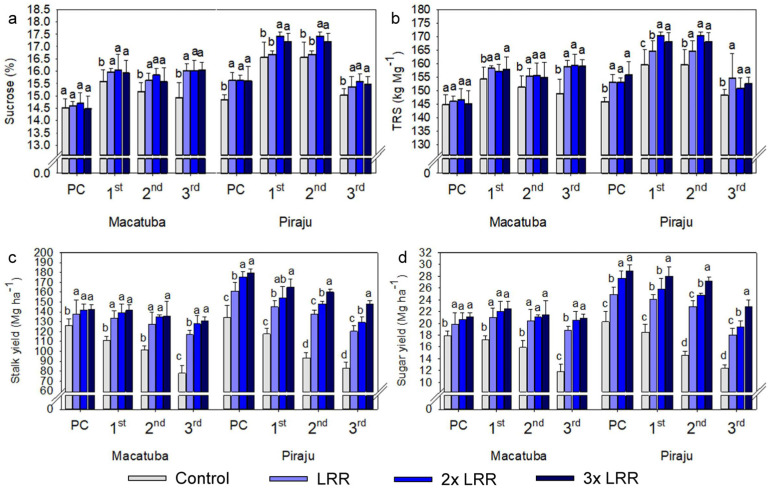
Sucrose (**a**), total recoverable sugar (TRS, **b**), and yields of stalks (**c**) and sugar (**d**) under increasing lime rates at Macatuba and Piraju under modified deep strip-tillage (MDT) in plant cane and first, second, and third ratoons. Different lowercase letters indicate significant differences between lime rates for each growing season (LSD, *p* ≤ 0.1). The error bars express the standard error of the mean (*n* = 4).

**Figure 7 plants-11-02110-f007:**
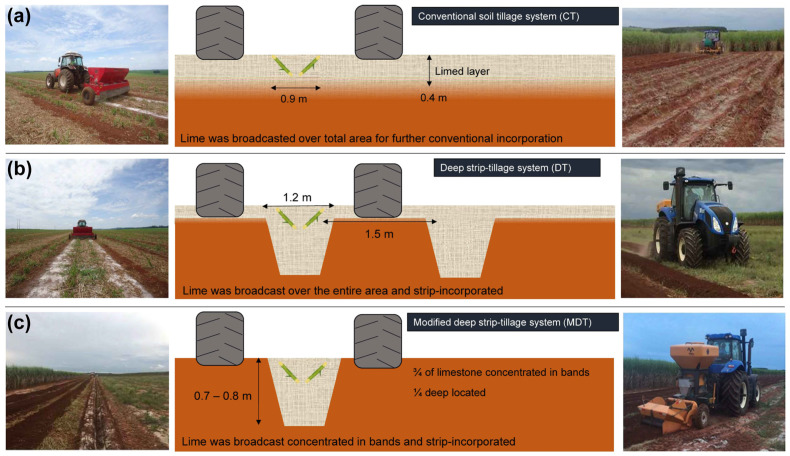
Illustration of the soil tillage systems; (**a**) conventional tillage system, (**b**) deep-strip tillage system, and (**c**) modified deep-strip tillage system, equipment, and the disposition of sugarcane plants.

**Table 1 plants-11-02110-t001:** Increments of stalk and sugar yields and agronomic efficiency index (AEI) values as a function of lime rate at Macatuba and Piraju.

Lime Rates	Stalks Yield						Sugar Yield					
	Increase (Mg ha^−1^) ^1^			AEI (%) ^2^			Increase (Mg ha^−1^)			AEI (%)		
	CT	DT	MDT	CT	DT	MDT	CT	DT	MDT	CT	DT	MDT
2016	Macatuba											
LRR	10.4	9.5	11.8	-	-	-	1.8	1.5	1.9	-	-	-
2× LRR	20.2	17.7	15.7	194	186	133	3.8	2.8	2.7	207	187	143
3× LRR	24.2	27.6	16.6	232	290	141	4.5	4.0	3.2	245	266	170
2017												
LRR	14.1	11.9	23.0	-	-	-	2.9	2.3	3.8	-	-	-
2× LRR	26.6	24.4	28.2	189	204	123	4.8	4.1	4.8	163	177	126
3× LRR	27.9	33.0	30.6	198	276	133	4.6	5.6	5.3	156	242	138
2018												
LRR	14.7	13.1	25.6	-	-	-	2.7	2.5	4.5	-	-	-
2× LRR	29.9	26.8	33.4	204	204	131	4.8	5.2	5.1	178	213	113
3× LRR	34.9	36.4	34.2	238	277	134	5.3	6.0	5.6	194	246	123
2019												
LRR	17.5	25.5	38.9	-	-	-	3.3	4.8	7.0	-	-	-
2× LRR	33.7	39.1	50.0	192	153	128	5.9	7.0	8.7	180	145	124
3× LRR	44.3	52.6	53.0	253	206	136	7.5	9.1	9.1	229	191	130
Complete ^3^												
LRR	14.2	15.0	24.8	-	-	-	2.7	2.8	4.3	-	-	-
2× LRR	27.6	27.0	31.8	195	180	128	4.8	4.8	5.3	180	173	124
3× LRR	32.8	37.4	33.6	232	249	135	5.5	6.2	5.8	203	224	135
2016	Piraju											
LRR	26.0	22.6	27.3				4.4	4.5	4.7			
2× LRR	29.2	25.4	41.2	112	113	151	5.2	4.9	7.4	119	110	159
3× LRR	28.5	27.0	45.3	110	120	166	6.0	5.5	8.7	137	124	186
2017												
LRR	29.5	26.3	27.7				5.7	4.9	5.6			
2× LRR	40.0	33.8	36.5	136	128	132	8.1	7.0	7.3	143	144	129
3× LRR	41.1	40.9	47.5	139	156	172	8.3	7.9	9.6	147	162	170
2018												
LRR	35.7	37.9	44.5				6.5	6.7	8.3			
2× LRR	39.4	44.1	55.1	111	116	124	7.8	8.5	10.2	120	128	123
3× LRR	52.0	54.7	67.0	146	144	150	9.9	10.0	12.6	152	150	152
2019												
LRR	20.9	33.0	37.4				3.4	5.6	5.7			
2× LRR	26.3	40.3	46.5	126	122	125	4.3	6.2	7.0	125	111	124
3× LRR	41.9	54.7	65.2	201	166	175	6.6	8.7	10.4	192	154	184
Complete												
LRR	28.0	29.9	34.2				5.0	5.4	6.1			
2× LRR	33.7	35.9	44.9	120	120	131	6.4	6.7	8.0	127	123	132
3× LRR	40.9	44.4	56.2	146	148	164	7.7	8.0	10.3	154	148	170

^1^ Increments in stalks and sugar yield relative to the control; ^2^ Agronomic efficiency index of two and three times the lime recommended rate (2 and 3× LRR) relative to the lime recommended rate (LRR) for stalk and sugar yields; ^3^ Complete sugarcane cycle (summed growing seasons).

**Table 2 plants-11-02110-t002:** Lime doses and forms of application under different soil tillage systems.

Soil Tillage System	CT ^1^	DT ^2^	MDT ^3^	
Lime Application	Broadcasted (over the Entire Area)	Broadcasted (over the Entire Area)	Broadcasted on Planting Furrow	Incorporated
Lime doses	Mg ha^−1^			
	Macatuba (Rhodic Hapludox)			
Control ^4^	0.0	0.0	0.0	0.0
LRR ^5^	5.0	5.0	3.8	1.2
2× LRR ^6^	10.0	10.0	7.5	2.5
3× LRR ^7^	15.0	15.0	11.3	3.7
	Piraju (Rhodic Hapludox)			
Control	0.0	0.0	0.0	0.0
LRR	4.0	4.0	3.0	1.0
2× LRR	8.0	8.0	6.0	2.0
3× LRR	12.0	12.0	9.0	3.0

^1^ Conventional soil tillage system; ^2^ Deep strip-tillage system; ^3^ Modified deep strip-tillage system; ^4^ No limestone application; ^5^ Limestone recommended rate [27]; ^6^ Two times limestone recommended rate; ^7^ Three times limestone recommended rate.

## Data Availability

Not applicable.

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
