# Peer review of "Lime Rate in Clayey Soils Influences Chemical Fertility and Sugarcane Yield"

_plants, 2022, doi:10.3390/plants11162110_

Round 1
Reviewer 1 Report
Good manuscript.
Please consider label all the subplots in Figure 7 as A, B, C, ..., and then, explain each of the subplots.
Author Response
The subplots were labeled in figure 7 and pointed out in the text.
Reviewer 2 Report
The Authors have improved what already was a remarkable manuscript, in my opinion. I mostly appreciate, in response to my doubts, the visual explanations of spatial lime distribution referred to soil incorporation and cane rows planting. Maybe this could have been incorporated into the mmanuscript, instead of being offered only to reviewers. In contrast to this, I cannot agree with the Authors' stubbordness in defending a statistical threshold of p = 0.1 instead of adopting the much more generally convened p = 0.05 (and p = 0.01). I cannot believe that at p = 0.05 most of the significant differences they found would disappear. It would be acceptable that the p = 0.1 be maintained in those cases where no significance/signficant difference were found at p = 0.05; in no other case.
However, it is up to them to support their work's scientific qualification also in statistical terms by following un-interested suggestions. If they do not want to, let this manuscript be published as is.
Author Response
We greatly appreciate the concern with the statistical threshold of p = 0.1. We tested p = 0.05 and in some cases it worked. However, in most cases the results did not express the reality. The authors' stubbornness in defending a statistical threshold of p = 0.1 is because LSD above 20 tons of cane (CV under 7-9%) does not express reality, unfortunately. We always try p = 0.05 or 0.01 before, but sometimes we can't use it. Considering sugarcane field studies, a threshold of p = 0.1 is highly acceptable for any scientific journal. We decided to keep the threshold of p = 0.1 to respect the significance of the differences.
Reviewer 3 Report
General comment: The significant changes from the original version of the manuscript, taking into account my comments as well, are satisfactory.
Author Response
We appreciate the comment.
This manuscript is a resubmission of an earlier submission. The following is a list of the peer review reports and author responses from that submission.
Round 1
Reviewer 1 Report
The reviewed paper concerns lime rate influences chemical fertility and sugarcane yield in clayey soils. The paper presents the parameters of chemical fertility with increasing doses of lime and in various tillage systems. Three experiments were conducted, from April 2015 to October 2019. The presented work is very extensive, the authors have shown the results in tables and figures. Figures and tables of results are also provided in the supplementary.
Comments:
1. The problem presented in the paper is important, but the result of the experience is well known. Please write what is new in the paper, explain why the authors undertook such research. 2. Please complete the chapter: 4.3. Soil chemical analysis. What apparatus was used - name, manufacturer. 3. Why was Statsoft Statistica 7.0 Software 2005 used for statistical analysis? There are new versions of the program. 4. Editorial errors are marked in yellow in the text.

Reviewer 2 Report
The paper is mainly addressed to sugarcane producers and shows the possibility of increasing sugar yields by changing the traditional soil liming system. The experiments carried out allowed to collect information which can be helpful in the proper management of sugarcane crops. Unfortunately, the way of preparing the manuscript causes many objections:
- large parts of the text are incomprehensible and contain many factually incorrect formulations. Perhaps this is due to the difficulty of translating long and complicated sentences by someone who is not an expert in this field?
-the structure of the article is not well thought out, e.g. in the description of the results there is information on doses and methods of liming application which should be included in the methodology
-lack of analytical presentation of research results, and only indication of generally known trends
- excessive detailing of technical aspects of the experiment (machinery used, sampling method) and marginalisation of the relevant research topic
-the lack of data on essential macronutrients (NPK), and the focus on Mg, Ca and micronutrients only, does not allow an assessment of the full change in soil chemical fertility under the studied factors
- the discussion is a literature review rather than a critical comparison of the results obtained with the current state of knowledge
In its current form, the manuscript does not meet the criteria for a scientific paper and needs to be fundamentally revised and substantive improvement.
Reviewer 3 Report
A generally good, well written work. The Authors have addressed the issue of liming acidic soils cultivated with sugar cane in sub-tropical Brazil. They have cross combined four lime doses including no lime, and three application methods: conventional tillage (0.3-0.4 m depth) to incorporate lime previously broadcasted on soil surface; deep strip tillage (0.6-0.7 m depth) in the same positions as subsequently planted sugar cane rows, with previously broadcasted lime; modified deep strip tillage, where lime will be partly broadcasted, partly deep placed (0.4 and 0.6 m depth) through the subsoiler. Soil chemical fertility (pH, CEC and its ionic components, and base and aluminum saturation) down to 1 m depth, sugar cane leaf nutrient concentrations (N, P, K, Ca, Mg, S, Fe, Cu, Zn, Mn, B) at peak growth, and final stalk yield, sucrose concentration, total recoverable sugars and sugar yield, in two sites across four years (planting year and three subsequent ratoons) in São Paulo, Brazil.
As a result, the work is very rich of data the Authors have been committed to describe and relate (Pearson's correlation and PCA) in order to get sound, generally valid results. Sometimes the work is a bit lengthy, but the statements made in the Results section always accurately describe the specific table/figure. The Discussion section is most interesting, as the Authors have been able to convey their apparently deep knowledge of the topic into very interesting considerations for the reader.
There is a series of minor points I wish the Authors address before the work is published (see attached pdf), but I have some remarks in the statistical analysis. First, they should justify why they chose a threshold for statistical significance at p = 0.1, while the generally accepted threshold is p = 0.05 (or p = 0.01). They also claim they used the t-test for separating > 2 means in statistically significant sources, which is unfeasible; they may have adopted the LSD or some other test. Lastly, in the (Pearson's? PLease, specify) correlations, they should justify why they chose a threshold for statistical significance (p = 0.1) at r = ±0.70. Such level is sufficienct for stating the statistical significance (p = 0.1) of a correlation betweem 7 paired values, while I think that their correlations were run between a much higher number of paired data (which they should better indicate), and a much lower r threshold could have been retained. In this respect, the adoption of a weak threshold for statistical significance (0.1) in all statistcal analysis is questonable, as the dataset could support stronger thresholds without losing much of its relevance.
Overall, I wish that these remarks be addressed and improved, for the work to be published soon.

Reviewer 4 Report
Good work!
Just a suggestion, linear-plateau model might be useful to gain the critical levels of soil pH, Ca, and Mg.